Construction and clinical application of a risk model based on N6-methyladenosine regulators for colorectal cancer

Zhu Hanhan 1 zhuhanhan321@163.com
Yang Yu 1
Zhou Zhenfeng 2
1 Oncology Department, The Sixth Affiliated Hospital of Jinan University , Dongguan , China
2 Cancer Diagnosis and Treatment Research Center, The First Affiliated Hospital of Jinan University , Guangzhou , China
Guan Fanglin
Electronic publication date: 2024 Dec 20
Publication date: 2024
Volume: 12
Electronic Location ID: e18719
Received 2024 Sep 19; Accepted 2024 Nov 26
Copyright: © 2024 Zhu et al.
Copyright year: 2024
Copyright holder: Zhu et al.
License: This is an open access article distributed under the terms of the Creative Commons Attribution License, which permits unrestricted use, distribution, reproduction and adaptation in any medium and for any purpose provided that it is properly attributed. For attribution, the original author(s), title, publication source (PeerJ) and either DOI or URL of the article must be cited.
License URL: https://creativecommons.org/licenses/by/4.0/

Keywords: Colorectal cancer (CRC), m6A regulators signature, RiskScore, Single cell analysis, Immune infiltration analysis

Funding: The authors received no funding for this work.

==============================
Background

Colorectal cancer (CRC) shows a high incidence in developed countries. This study established a prognosis signature based on N6-methyladenosine (m6A) regulators involved in CRC progression.

Method

The bulk RNA-seq data from the Atlas and Compass of Immune-Colon cancer-Microbiome interactions (AC-ICAM) and GSE33113 CRC datasets were obtained from the cBioportal and GEO databases, and a total of 21 m6A regulators genes were collected from a previous study. The scRNA-seq analysis of the GSE146771 cohort was conducted applying the Seurat and harmony R packages. Consensus clustering based on the expressions of m6A regulators was performed with the ConsensusClusterPlus package. The ggGSEA package was used for the Gene Set Enrichment Analysis (GSEA). The un/multivariate and LASSO Cox analysis were performed applying the “survival” and “glmnet” packages for developing a risk model. The pRRophetic and GSVA packages were utilized to analyze potential drugs for CRC and immune infiltration in different risk groups, followed by the Kaplan–Meier (KM) survival and ROC analysis with the “survival” and “timeROC” packages. In vitro assays included the quantitative polymerase chain reaction (qPCR), wound healing and transwell were performed.

Results

CRC patients in the AC-ICAM cohort were assigned into three molecular subtypes (S1, 2 and 3) based on nine m6A regulator genes. Specifically, the prognostic outcome of the S3 was the most favorable, while that of the S1 was the worst and this subtype was associated with the activation of NF-kB, TNF-α and hypoxia pathways. Three key genes, namely, methyltransferase-like 3 (METTL3), insulinolike Growth Factor2 mRNA-Binding Protein 3 (IGF2BP3) and YTH domain-containing protein 2 (YTHDC2), selected from the 9 m6A regulator genes were combined into a RiskScore, which showed a high classification effectiveness in dividing the patients into high- and low-risk groups. Inhibition of the expression of METTL3A or that of IGF2BP3 suppressed the invasion and migration of CRC cells. Notably, the high-risk patients had higher immune cell infiltration to support the activation of multiple immune responses and exhibited significantly poor prognosis. Meanwhile, a nomogram with practical clinical value was developed based on the RiskScore and other clinical features. Finally, eight potential drugs associated with the RiskScore were identified, and CD4+ cells and Tregs were found to be closely associated with CRC progression.

Conclusion

The RiskScore model developed based on m6A regulators played a critical role in CRC development and can be considered as a prognosis predictor for patients with the cancer. The present discoveries will facilitate the diagnosis and clinical management of CRC patients.

Introduction

Colorectal cancer (CRC) shows a high incidence all over the world (Bou Malhab & Abdel-Rahman, 2022). The latest statistics showed that CRC is the third most common cancer (Siegel & Giaquinto, 2024), with estimated new cases of 152,810 and estimated deaths of 53,010 in 2024 (Keum & Giovannucci, 2019). A high possibility of metastasis to the liver, lungs, and peritoneum will noticeably increase the death rate of patients suffering from CRC (Tang et al., 2021). At present, nation-wide screening programs, use of colonoscopy (Bao et al., 2020), early interventions and treatment options have greatly lowered the mortality of CRC by approximately 35% from 1990 to 2007 (Siegel et al., 2017), however, the 5-year survival of CRC patients with metastasis is lower than 12% (Siegel & Miller, 2019). Thus, to investigate the molecular mechanisms of CRC progression and to identify novel biomarkers may help improve the survival of CRC patients.

Epigenetic modifications not involved in changing nucleotide sequence could affect gene expression and phenotype (Zinovkina et al., 2024). Currently, multiple epigenetic pathways such as chromosome remodeling, DNA methylation, histone modification and non-coding RNA regulation have been found (Zhao et al., 2020). The RNA-level modification includes N1-methyladenosine, 5-methylcytosine, pseudouridine, and N6, 2′-O-dimethyladenosine (m6A) (Jonkhout et al., 2017), among which m6A modification accounts for approximately 60% of all RNA modifications and affects almost all RNA metabolic activities including translation, transport, splicing and degradation (Batista et al., 2014). The modification of RNA methylation represents a dynamic and reversible process in which m6A methyl-transferases, m6A binding proteins, and demethylases act as ‘writers’, ‘readers’, and ‘erasers’ at the m6A site, respectively (Zhuang et al., 2020). A previous study demonstrated that aberrant regulators in m6A RNA methylation are associated with many diseases including tumorigenesis process through significantly affecting gene expressions (Lin et al., 2016). For example, the abundance of m6A regulated by abnormal expressions of m6A regulators could disrupt the expressions of oncogenes, thereby promoting tumorigenesis and metastasis (Huang et al., 2018; Liu et al., 2020). Chen et al. (2019) reported that the m6A RNA methylation regulators promote the malignant development, acting as a prognosis marker of bladder cancer. Ma et al. (2019) also revealed the vital interaction relationship between non-coding RNAs and m6A RNA methylation in cancer. Recent studies have confirmed m6A regulators as promising biomarkers for cancer diagnosis and intervention (Huang et al., 2016; Chen et al., 2018) and their potential of serving as therapeutic targets in cancer treatment (Zhang & Xin, 2018).

Based on the RNA sequencing data from CRC patients, we identified m6A regulatory factors closely involved in shaping the prognostic outcomes of CRC and classified patients into different molecular subtypes to observe the prognostic differences among subtypes and their related signaling pathways. In addition, we established a RiskScore based on the key m6A regulators to help stratify the survival risk of CRC patients, and also explored potential therapeutic targets with immune infiltration and drug sensitivity analyses. This study discovered new biomarkers for individualized prognostic assessment for CRC patients and revealed features associated with m6A modifications, providing theoretical support for precision medicine treatments and immunotherapy in CRC.

Materials and Methods

Data collection

The AC-ICAM cohort containing the RNA-Seq expression and clinical follow-up data of disease-specific survival (DSS) of 320 CRC patients was downloaded from the cBioportal database (https://www.cbioportal.org/; Roelands et al., 2023). The GSE33113 cohort containing 90 CRC samples and a single-cell RNA-seq cohort (GSE146771) were downloaded from the Gene Expression Omnibus (GEO, https://www.ncbi.nlm.nih.gov/geo/) (Song et al., 2023). A gene set of 21 m6A regulator genes was collected from a previous study (Li et al., 2022). Significant prognostic genes were selected by univariate Cox regression analysis.

Single-cell data preprocessing

The scRNA-seq analysis was conducted utilizing the Seurat R package (Zulibiya et al., 2023). For quality control of the raw data, the PercentageFeatureSet package was utilized to filter cells expressing at least 200 to 4,000 genes that were present in more than three cells with the mitochondrial gene content <5%. After normalizing the data of 10 samples using SCTransform function, highly variable genes (HVGs) were screened by the FindVariableFeatures. Next, principal component analysis (PCA) was performed to identify anchor point based on the expression of HVGs (setting dim = 20) after normalizing the scale with the ScaleData (Lopes & Tenreiro Machado, 2021). Then, batch effect between samples was removed by the harmony package (Korsunsky et al., 2019). After performing dimensionality reduction with the RunTSNE function, the FindNeighbors and FindClusters function was used for the clustering of Cell subsets at a resolution of 0.1. The CellMarker2.0 database provided marker genes for cell type annotation.

Identifying the molecular subtypes based on m6A regulators

According to the expressions of m6A regulator genes, the “HC” algorithm with “pearson” as the metric distance was used to conduct consensus clustering analysis in the ConsensusClusterPlus R package (Wilkerson & Hayes, 2010), and 500 bootstraps (each bootstrap process included 80% patients) was executed. The optimal number of clusters was determined in a range between 2 and 10, according to consensus cumulative distribution function (CDF).

Gene set enrichment analysis (GSEA)

The gene set of cancer HALLMARK pathways was downloaded from the Molecular Signatures Database (MSigDB, https://www.gsea-msigdb.org/gsea/msigdb/; Liberzon et al., 2015) to analyze potential signaling pathways and functional differences among various cell clusters using the ggGSEA R package. Pathways with a false discovery rate (FDR) <0.05 were defined as significantly enriched. The Pathway RespOnsive GENes (PROGENy) algorithm was used to calculate the activities of cancer-related pathways for each sample (Malta et al., 2018).

Construction of risk prognosis model

Candidate m6A regulators for developing a risk model was refined by LASSO Cox regression analysis using the glmnet R package (Chu et al., 2023), and multivariate Cox regression analysis was used for calculating regression coefficient. A RiskScore was constructed based on the formula: RiskScore =Σβi×Expi, where i represented the expression of risk model genes and β was the regression coefficient. The RiskScore of patients was to divide the patients into different risk groups based on the median RiskScore value. Prognostic differences among different patients were compared by performing Kaplan-Meier (KM) survival analysis with the logarithmic rank test using the survival R package (Chu et al., 2023). The timeROC R package was use for the receiver operating characteristic (ROC) analysis to assess the classification performance of the model (Chu et al., 2023).

Correlation analysis between drug sensitivity and risk score

The IC50 for patients in the AC-ICAM cohort was predicted by the pRRophetic R package, followed by performing correlation analysis between patients’ drug sensitivity and RiskScore using the spearman algorithm to select drugs with p < 0.05 & | cor | > 0.3 (Geeleher, Cox & Huang, 2014). Tumor-infiltrating lymphocytes (TILs) score of 28 immune cells was calculated with the GSVA package and the marker genes identified by a previous article (Charoentong et al., 2017).

Cell culture and qPCR assay

Dulbecco’s Modified Eagle Medium (DMEM) containing 1% penicillin-streptomycin (GIBCO, USA) and 10% fetal bovine serum (FBS, Gibco, CA, USA) was used for culturing human colorectal adenocarcinoma cells (SW116, CVCL_0544) and human normal intestinal epithelial cells (NCM460) purchased from the American Type Culture Collection (ATCC, Manassas, VA, USA). The cells were incubated at 37 °C in 5% CO2. Total RNA was separated by the Trizol reagent (Vazyme, Nanjing, China), and 50 ng of total RNA was used for synthesizing cDNA applying the Vazyme Reverse Transcription Kit (Vazyme, Nanjing, China). Then, the SYBR qPCR Master Mix (Vazyme, Q511–02) was applied for the qPCR detection according to the specification (Sindhuja, Amuthalakshmi & Nalini, 2022). The expressions of genes were calculated with 2–ΔΔCT method, with β-actin as a normalized reference. The sequences of specific primer were as follows: METTL3 (F: 5′-ATTAGACAAAAATAAGGAAGAAATTGCC-3′, R: 5′-AATTATCTAGGTCCTATATAGCCATAAAGG-3′), YTHDC2 (F: 5′-TCCAGAACAAGTAAGAGGAGTCGTT-3′, R: 5′-CACACAGATACAAAATATGAAAAACATACA-3′), IGF2BP3 (F: 5′-CTGTGGGGACCGCGGCTT-3′, R: 5′-ACGGTCGGAGGGGTCGAC-3′) and β-actin (F: 5′-CCTCGCCTTTGCCGATCC-3′, R: 5′-GGATCTTCAT GAGGTAGTCAGTC-3′).

Wound healing and transwell assay

We used the si-METTL3 regent (sense: UGUUUAUUGAUAAUUCGUCUG and antisense: GACGAAUUAUCAAUAAACACA) and IGF2BP3 regent (sense: CAGUAUAACAGAUAUUCUAAU and antisense: UAGAAUAUCUGUUAUACUGUG) (Sangon, Shanghai, China) to silence METTL3 and IGF2BP3 (Zhao et al., 2023). SW1116 was transfected into cells by Lipofectamine 3000 (Invitrogen, Waltham, MA, USA; L3000015). YTHDC2 was overexpressed by the pcDNA3.1 vector (Thermo Fisher Scientific, Waltham, MA, USA). To measure cell migration with wound healing assay, 6-well plates (Corning, Glendale, AZ, USA) containing DMEM were seeded with 4 × 106 cells to incubate the cells to confluency, and a rectilinear scratch was created using a 20-μL pipette tip. After 48 h, the cells were washed by PBS and photographed and counted under an inverted microscope (Carl Zeiss, Oberkochen, Germany) (Liang et al., 2023). Cell invasion was measured by transwell assay. First, a 24-well plate (Corning) with 8-μm pore inserts was used for cell culture, with in the upper chamber well containing 4 × 104 cells in 200 μL serum-free DMEM and the lower chamber containing 800 μL of DMEM and 20% FBS. After 48 h, the cells in lower chamber were fixed by 4% paraformaldehyde and stained by 0.1% crystal violet for 15 minutes (min), respectively. The cells were photographed under an inverted microscope (Carl Zeiss, Oberkochen, Germany) (Liang et al., 2023).

Statistical analysis

The R software (version 3.6.0) was employed for statistical analysis and data visualization. Correlation analyses were performed based on the Pearson method while ensuring that the data conformed to a normal distribution. Log-rank test was used to compare differences between the survival curves of patients in different risk groups. For gene-specific validation, we used t-test to compare significant differences between groups. p < 0.05 was considered statistically significant (*p < 0.05, **p < 0.01, ***p < 0.001, ****p < 0.0001).

Results

Classification of three molecular subtypes with different prognosis based on m6A regulator genes

Univariate Cox regression analysis showed that among the 21 m6A regulator genes in the AC-ICAM cohort, nine genes such as METTL3 and METTL14 were significantly correlated with patients’ prognosis (p < 0.05, Fig. 1A). According to the CDF of consensus clustering analysis, the patients were clustered into three molecular subtypes (S1, S2 and S3) by the expressions of the 9 m6A regulators genes, with the S3 having a better prognosis and the S1 exhibiting significantly worse outcome (Figs. 1B and 1C). Analysis of the expression differences of the nine genes demonstrated that IGF2BP3, WTAP and METTL3 were significantly overexpressed in the S1 subtype, while YTHDC2, HNRNPA2B1, RBM15B and YTHDF1 were significantly upregulated in the S3 subtype (p < 0.05, Fig. 1D). PCA analysis also revealed a clear boundary between the three subtypes (Fig. 1E). Next, we compared the distribution proportion of the three subtypes in different clinical features and observed that most patients in S1 subtype had a higher clinical grade as a higher proportion of S1 patients with T4 stage, N2 stage, M1 stage and Stage 4 were found (Fig. S1A). These results suggested that the m6A regulator signatures could be an effective and reliable prognostic phenotype.

Figure 1 Molecular classification based on m6A regulators.

(A) Forest plot of m6A regulators genes associated with prognosis after univariate Cox regression analysis. (B) Prognostic differences among three subtypes in AC-ICAM cohort. (C) Heatmap of consensus clustering analysis. (D) The heatmap of 9 m6A regulators genes expression among the three subtypes. (E) PCA analysis of three molecular subtypes.

Patients in the three molecular subtypes had different pathway activation

Difference of activated pathways between the three subtypes in the AC-ICAM cohort was analyzed by GSEA. As shown in Fig. 2A, inflammatory response and cell cycle-related pathways such as oxidative phosphorylation, DNA repair, TNFα signaling via NFκB, allograft rejection and Myc-targets-v1 were significantly activated in S1 subtype, while tumor-related pathways of Mitotic spindle, Hedgehog signaling, epithelial mesenchymal transition (EMT), and Wnt-β-catenin signaling were activated in S3 subtype. Next, we calculated the carcinogenic pathway activity score based on one-way analysis of variance (ANOVA), and found that the S1 subtype had significantly higher activity of NF-kB, TNF-α, hypoxia, and TRAIL pathway activation (p < 0.05, Fig. 2B). Finally, comparison of the immunologic constant of rejection (ICR) score of the three subtypes (Sherif et al., 2022) showed that the S1 subtype had significantly ICR score (p < 0.05, Fig. 2C). The above results indicated that various signaling pathways involved in affecting the prognosis of CRC patients in the three molecular subtypes were activated.

Figure 2 Pathway characteristics analysis between three molecular subtypes.

(A) GSEA analysis of three molecular subtypes in AC-ICAM cohort. (B) Differences in carcinogenic pathway activity among three subtypes. ANOVA was used to compare the overall differences between the three subtypes (S1, S2 and S3). (C) ICR score differences among three subtypes (*p < 0.05, ***p < 0.001, and ****p < 0.0001).

Development of a risk prognostic model and validation

Least absolute shrinkage and selection operator (LASSO) Cox with stepwise regression analysis was used to select model genes from the nine prognosis-related m6A regulator genes. Figure 3A displayed the coefficient changes of different m6A-regulated genes at different Lambda values, and it could be observed that the model with three genes was the optimal and relatively stable when λ was 2 (Fig. 3B). Next, the regression coefficient (Fig. 3C) for the risk model was calculated by the multivariate Cox regression analysis as RiskScore =0.559∗METTL3+(−1.327)∗YTHDC2+0.125∗IGF2BP3. Each patient was assigned with a RiskScore, and the ROC analysis showed an area under curve (AUC) value of 0.67, 0.67, 0.67, 0.7 and 0.66 for 1-, 2-, 3-, 4- and 5-year overall survival (OS), respectively (Fig. 3D), indicating that the model was highly accurate in long- and short-term classification. After dividing the patients into high- and low-risk groups by the median RiskScore value, the KM survival analysis revealed that the high-risk patients in the AC-ICAM cohort had remarkably overexpressed METTL3 and IGF2BP3 and significantly poor prognosis (p < 0.05, Fig. 3E) (Fig. 3F). Moreover, we used the GSE33113 cohort as validation set to analyze 1–5 year survival rate, prognostic differences and the expressions of the model genes between high- and low-risk groups. Similarly, the AUC value of 1–5 year survival were all higher than 0.65, and the high-risk patients also had significantly high-expressed METTL3 and IGF2BP3 and poor outcomes (Figs. 4A–4C).

Figure 3 Establishment of clinical prognostic model.

(A) LASSO coefficient trajectory plot. (B) LASSO regularized trajectory plot. (C) Forest plot of model gene after multivariate Cox regression analysis. (D) ROC curve of RiskScore at 1–5 years OS in AC-ICAM cohort. (E) KM survival analysis among high and low risk patients. (F) Heatmap of model genes expression among high and low groups.

Figure 4 Validation of clinical prognostic model.

(A) ROC curve of RiskScore at 1–5 years OS in GSE33113 cohort. (B) KM survival analysis among high and low risk patients in GSE33113 cohort. (C) Heatmap of model genes expression among high and low groups in GSE33113 cohort. (D) The difference analysis of RiskScore in various clinicopathologic features.

RiskScore model was an independent factor for CRC prognosis and the development of a nomogram

Further comparison on the clinicopathological differences between the two risk groups in the AC-ICAM cohort demonstrated that gender and path.metastasis.Stage of the two risk groups had no significantly difference but features such as age, path.tumor.Stage, path.nodes.Stage, and AJCC.path.Stage were significantly different. Additionally, the RiskScore increased with the clinical grades (p < 0.05, Fig. 4D). After incorporating the RiskScore and these clinicopathological features, univariate and multivariate Cox regression analysis showed that the RiskScore, path.nodes.Stage1 and path.metastasis.Stage M1 were significant independent prognostic factors for CRC (p < 0.05, Figs 5A, 5B). To estimate the survival for CRC patients, we combined Node.Stage, Metastasis.Stage and Riskscore to establish a nomogram model (Fig. 5C). According to the calibration curve, the predicted survival rates for 1, 3 and 5 year(s) were close to the standard curve (Fig. 5D), indicating that the prediction of the nomogram was highly accurate. Moreover, the decision curve (DCA) demonstrated that the net benefit of nomogram model was significantly higher than the extreme curve (Fig. 5E), suggesting the nomogram model had clinical application probability.

Figure 5 Identifying independent prognostic factor and constructing a nomogram.

(A) Univariate Cox regression analysis for the significant prognostic factor. (B) Multivariate Cox regression analysis for independent prognostic factors. (C) A developing nomogram. (D) Calibration curve of nomogram. (E) Decision curve of nomogram.

Identifying eight potential drugs associated with the RiskScore and differences in activated pathways

The ssGSEA method was employed to assess the differences of immune infiltration between high- and low-risk patients. The results showed that γδ T cells, neutrophils, MDSC, activated CD4 T cells, natural killer T cells, immature dendritic cells, activated CD8 T cells, CD56dim natural killer cells, mast cells, activated dendritic cells, macrophages, Type 1 and 2 T helper cells had significantly higher infiltration levels in high-risk patients, while the low-risk patients had the higher infiltration of memory B cells and activated B cells (Fig. 6A). The pRRophetic R package was used to predict the drug sensitivity of low- and high-risk patients, and eight potential drugs closely associated with the RiskScore (p < 0.05 & | cor | > 0.3) were screened, including BMS-754807, Lisitinib and LFM-A13 (Fig. 6B). The GSEA enrichment analysis showed that inflammatory pathways such as interferon_gamma_response, inflammatory_response, interferon_alpha_response were activated in the high-risk group. Additionally, some pathways including mitotic spindle, UV response, and estrogen response early were suppressed in the high-risk patients (Fig. 6C).

Figure 6 Difference of immune microenvironment of two risk groups and the drug sensitivity analysis.

(A) Immune infiltration difference among the high and low risk groups. (B) Correlation analysis between the RiskScore and its potential drugs. (C) The GSEA analysis among the high and low risk groups. (*p < 0.05, **p < 0.01, ***p < 0.001, and ****p < 0.0001).

Differences in the expression distribution and activity of m6A regulatory factors in different cell types at the single-cell level

The GSE146771 cohort was used for scRNA-seq analysis to study cancer-associated cells at the single-cell level. After quality control of raw data, 10,186 cells were obtained and divided into 11 types based on the expressions of markers, such as B cells 1 and 2, CD4+ cells, NKT cells, CD8+ T cells, mast cells, fibroblasts, epithelial cells, monocytes, Tregs cells, neutrophils (Figs. 7A, 7B). It was found that YTHDC2 was high-expressed in B cells and METTL3 was high-expressed in CD4+ cells and IGF2BP3 expression was not observed in the above cells (Fig. 7C). Then, the AUCell method was used to calculate the m6A regulator score of each cell, and we found that the Tregs cells had the highest m6A regulator score, and that there were significant differences in m6A regulator score among different cells (p < 0.05, Fig. 7D).

Figure 7 Single cell RNA-seq analysis.

(A) The landscape of the identified cell clusters. (B) The marker genes expression of each cell type. (C) The model genes expression in various cell type. (D) The difference of m6A regulators score in various cell type.

The expressions of model genes and migration and invasion assay in vitro

Finally, we measured the expressions of the three model genes and analyzed their potential functions in vitro. The results of qPCR showed that METTL3 and IGF2BP3 were significantly overexpressed in the tumor cells, while YTHDC2 was significantly downregulated (p < 0.05, Fig. 8A). In addition, the wound closure rate of SW1116 cells in the si-METTL3 and si-IGF2BP3 groups was noticeably lowered than the control group (p < 0.05, Figs. 8B, 8F). At the same time, after silencing of METTL3 and IGF2BP3, the number of the migration cells was also significantly reduced in the si-METTL3 and si-IGF2BP3 group (p < 0.05, Figs. 8C, 8G). However, we observed that overexpression of YTHDC2 did not affect the migration and invasion of CRC cells (p > 0.05, Figs. 8D, 8E). This suggested that METTL3 and IGF2BP3 were important pro-carcinogenic factors in CRC progression, showing a potential value to be used for targeted therapy. However, the role of YTHDC2 remained to be further investigated.

Figure 8 qPCR and wound healing and trans-well assay.

(A) qPCR for the expression of three model genes (METTL3, IGF2BP3 and YTHDC2). (B and C) Knockdown of METTL3 in SW1116 cells reduces the cellular wound healing rate (B) and the number of invading cells (C). (D and E) Overexpression of YTHDC2 did not significantly affect the migration (D) and invasion (E) ability of SW1116 cells. (F and G) Knockdown of IGF2BP3 significantly inhibited the wound healing rate (F) and the number of invading cells (G) in SW1116 cells All experimental data of independent triplicates were expressed as mean ± standard deviation. (**p < 0.01, ***p < 0.001, and ****p < 0.0001).

Discussion

CRC is a primary cause of cancer mortality in the world (Sharma et al., 2024; Bao et al., 2020). Despite the development of therapeutic strategies and diagnostic methods for CRC, the clinical outcome of CRC patients is still unsatisfactory due to rapid progression, early metastasis and advanced stage (Hissong & Pittman, 2020). M6A is a vital form of epigenetic modification involved in carcinogenesis of various cancers when aberrantly expressed, and various m6A regulators fulfill distinctly different functions in different cancers (Deng et al., 2018). This study evaluated the prognostic correlation of m6A phenotype in CRC, in which nine m6A regulators associated with the prognosis of the cancer can divide the patients into different molecular subtypes with significant differences in pathway activation and prognosis. Based on these nine m6A regulators, we constructed a RiskScore and a nomogram to facilitate the prognostic evaluation for CRC patients and also analyzed immune infiltration differences among high- and low-risk groups of patients. It was found that the high-risk group had higher infiltration of immune cells such as γδ T cells, macrophages, activated CD4 T cells, mast cells and activated CD8 T cells and stronger activation of inflammation pathways such as interferon_gamma_response and inflammatory_response. These results indicated that our RiskScore incorporated features specific for the prognostic prediction in CRC.

The CRC patients were divided into three molecular subtypes (S1, S2 and S3) with different prognosis. Specifically, the S1 subtype had a poor outcome and higher clinical grade and significantly activated inflammatory response and cell cycle pathways. Inflammation is a hallmark of cancer progression (Hanahan & Weinberg, 2011) and tumor-related inflammation is mainly related to local immune reaction at tumor site, which could greatly contribute to cancer progression (Gonzalez, Hagerling & Werb, 2018). Tumor will release a series of cytokines and inflammatory factors to recruit immune cells to the tumor site, leading to the production of most tumor-associated immune cells such as T-depletion cells and tumor-associated macrophages (TAMs) that support tumor growth and invasion under chronic inflammation condition (Khandia & Munjal, 2020). Activated tumor necrosis factors (TNFs) pathway in the S1 group is a marker for the poor prognosis of many malignancies, for example, CRC, lung and breast cancers (Richards et al., 2011). Activated NF-kB, hypoxia and TRAIL pathway in the S1 group mediate stress and cytokines responses, and aberrant activation of these signaling is associated with cancer, inflammation and immature immunity (Korbecki et al., 2021). These findings suggested that chronic inflammation exacerbates cancer progression and contributes to a poor prognosis in CRC.

Among the three model genes, METTL3 has been defined as a “writer”, while IGF2BP3 and YTHDC2 have been regarded as “readers”. Multivariate Cox regression analysis showed that METTL3 and IGF2BP3 were risk factors, whereas YTHDC2 was a protective factor. METTL3 is a key catalytic subunit of methyltransferase complex for m6A modification. Li et al. (2019) reported that the METTL3 is a high-overexpressed oncogene in CRC metastatic tissues, and that knockdown of METTL3 remarkably inhibited the self-renewal of stem cells and migration of tumor cells. Zhu et al. (2020) revealed that METTL3 enhanced the proliferation of CRC cells through methylating m6A site and stabilizing CCNE1. We also demonstrated that METTL3 was overexpressed in the tumor cells to promote the migration and invasion of CRC. Additionally, programmed death ligand-1 (PD-L1) suppresses T-cell function by binding to PD-1 receptor. IGF2BP3 can activate PD-L1 mRNA to regulate METTL3 to inhibit tumor immune surveillance (Wan et al., 2022). YTHDC2 is another important m6A regulator. Liu et al. (2022) found that significantly downregulated YTHDC2 in CRC tissues is closely related to worse disease-free survival and poor overall survival, and that downregulated YTHDC2 enhances the chemoresistance of LIMK1-mediated m6A-RNA methylation in CRC (Chen et al., 2023). These previous studies were consistent with our findings.

The immune infiltration analysis revealed that most immune cells such as γδ T cells, activated CD4 T cells, macrophages, activated CD8 T cells, and mast cells were more abundant in the high-risk group, while the low-risk group had the higher infiltration level of memory B cells. The infiltration of immune cells in most solid tumors is either a form of antitumor response (immunosurveillance) or subversion of immune system that facilitates tumor escape (Grivennikov & Karin, 2010). Although these immune cells in the tumor microenvironment (TME) represents the results of an active recruitment of immune cells to generate antitumor response, the presence of these immune cells including the recruitment of inhibitory cytotoxic immune-cells could also be repolarized for pro-tumorigenesis role (Grivennikov & Karin, 2010). Preclinical studies demonstrated that the TME plays a vital role in ultimately determining the anti-tumor or pro-tumor properties of immune cells, for example, in some TME the presence of conspicuous infiltration of inflammatory cells (particularly T lymphocytes) in the cancer nests or invasive margin is associated with a better prognosis (Galon et al., 2006). CRC patients without tumor necrosis or microsatellite instability have higher T-cell infiltration (Richards et al., 2012). Thus, it is speculated that tumor and its TME will influence immune and inflammatory infiltrate to produce anti-cancer response and favorable clinical outcomes in certain contexts. However, the cytokines that support the growth and invasion of tumor could also change the phenotype of the recruited immune cells. These immune cells, for example, neutrophils, macrophages and MDSCs are commonly associated with tumor progression and poor prognosis (Richards et al., 2012). The above results revealed that the immune activity of high-risk patients is higher, but inhibitors targeting macrophages are equally necessary in anti-cancer treatment. Our study validated the prognostic value of m6A-regulators in CRC, however, some limitations should also be pointed out. Firstly, this was a retrospective study and our findings need to be validated independently in larger CRC cohorts. Secondly, biases might exist in our model as the inclusion criteria, annotated types, and numbers of RNAs of the cohorts analyzed in this study were all different.

Conclusion

In summary, we constructed a RiskScore model for patients with CRC based on m6A regulators. The high-risk group showed significant activation of pro-inflammatory response pathways. This study found that METTL3 and IGF2BP3 facilitated the migration and invasion of CRC cells but the role of YTHDC2 remained to be explored by further studies. Our study provided potential molecular targets and a theoretical basis for the development of immunotherapy and targeted therapeutic strategies for CRC patients.

Supplemental Information

Supplemental Information 1 Clinical difference analysis.

(A) Differences in clinical characteristics of three subtypes in AC-ICAM cohort. (*p<0.05, **p<0.01, ***p<0.001)

Abbreviations

CRC Colorectal cancer

GSEA Gene Set Enrichment Analysis

m6A N6, 2′-O-dimethyladenosine

METTL3 Methyltransferase-like 3

LASSO least absolute shrinkage and selection operator

OS overall survival

DSS disease-specific survival

GEO Gene Expression Omnibus

HVGs highly variable genes

PCA Principal Component Analysis

CDF cumulative distribution function

MSD Molecular Signatures Database

MET mesenchymal-epithelial transition

ICR immunologic constant of rejection

Additional Information and Declarations

Competing Interests

Author Contributions

Data Availability

The authors declare that they have no competing interests.

Hanhan Zhu conceived and designed the experiments, performed the experiments, analyzed the data, prepared figures and/or tables, authored or reviewed drafts of the article, and approved the final draft.

Yu Yang conceived and designed the experiments, analyzed the data, authored or reviewed drafts of the article, and approved the final draft.

Zhenfeng Zhou performed the experiments, analyzed the data, prepared figures and/or tables, authored or reviewed drafts of the article, and approved the final draft.

The following information was supplied regarding data availability:

The datasets are available at NCBI: GSE33113 and GSE146771.

The raw data is available in GitHub, Zenodo, and Figshare:

- https://github.com/zhuhanhan1/Raw-data.git

- zhuhanhan1. (2024). zhuhanhan1/Raw-data: 1.1.2 Updated raw data (v.1.1.2). Zenodo. https://doi.org/10.5281/zenodo.14049389

- Zhu, Hanhan; Yang, Yu; Zhou, Zhenfeng (2024). origin_datas.zip. figshare. Dataset.

https://doi.org/10.6084/m9.figshare.26968165.v3.

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
