# Peer review of "Construction and clinical application of a risk model based on N6-methyladenosine regulators for colorectal cancer"

_PeerJ, doi:10.7717/peerj.18719_

## Round 0.1 · original submission · Major Revisions

While one reviewer suggested minor revisions, the other reviewer (r2) recommended major revisions. After thoroughly evaluating both sets of comments, I concur that major revisions are necessary to address all the concerns raised and to significantly improve the quality and impact of your work. Please carefully address all the points raised by both reviewers in your revision. Pay particular attention to the more substantial concerns highlighted by the reviewer recommending major revisions.

Reviewer 1 ·

Basic reporting

See additional comments

Experimental design

See additional comments

Validity of the findings

See additional comments

Additional comments

In this study, the author collected 21 m6A regulators and explored their relationship with the Colorectal cancer (CRC) progression. Then, based on the significant prognostic m6A regulators and the Cox regression analysis algorithm, a RiskScore model for risk stratification and prognosis, immune infiltration assessment was constructed, the model showed some superiority in predicting the survival probability of patients, this study has potential significance for improving the clinical treatment decision of CRC. However, the manuscript required further refinement before publication.
1. Line 47-48, the significance of this article should be emphasized in here.
2. Line 64-65, the early interventions and optimal treatment options decreased the mortality of patients. Current, what effective methods have been applied to the treatment of CRC.
3. Line 83-94, the m6A regulators is a type of epigenetic modification, it affected the progression of many types of cancer. The studies on m6A regulators affecting CRC development included what.
4. Line 201-212, the author preformed the consensus clustering analysis based on the prognostic genes expression and obtained 3 molecular phenotypes with different prognosis. This analysis does not seem to be closely related to the later content. So, what is the purpose of this analysis here.
5. Line 283-284, the author analyzed the expression of model genes in single cells, what is the purpose of this analysis, and the m6A regulators score difference in different cells implied what.
6. Line 278, The connection between title and content is not strong.
7. What is the specific clinical application of RiskScore and how does it improve patient outcomes.
8. Line 273-277, The GSEA enrichment analysis revealed some key pathways that activated in high risk patients, whether the specific regulation mechanisms of them on CRC progression were added into the discussion section.
9. The resolution of the Figure should be increased to meet magazine requirements.
10. What is the prospect of this article.

Reviewer 2 ·

Basic reporting

The process of gene screening in this study is experimentally rich, logically clear, and persuasive. However, the incomplete validation experiments make this study seem like a semi-finished product. It is recommended to supplement more experiments in order to obtain meaningful results.
(1) English proficiency: English needs polishing, such as "there reported", "high confidence", etc., which need to be modified to professional English.
(2) Reference accuracy: Some methodologies lack references, such as steps like Single cell data preprocessing.

Experimental design

(1) The supplier and catalog number of the cells need to be provided.
(2) Introduction: The content is lengthy and contains a lot of irrelevant information to this study, such as the description of the causes of colon cancer, which is related to increased smoking and alcohol consumption, high intake of fat and processed meat, lack of physical activity, and low intake of fiber containing green leafy vegetables. However, this article does not have any relevant experimental results. Authors from line52-line69 need to make some deletions and summarize, striving for a logical and concise writing of the paper.
In addition, there is no need to repeat the method described in this article (lines 95-107) in introduciton, only a brief explanation of the purpose of this article is needed.
(3) Method:
1) Writing standardization: Lack of information on the manufacturer, country, and item number of some instruments and reagents.
2) Statistical analysis: The description of statistical methods in this article lacks a description of statistical methods for scratch experiments and qPCR experiments. Please add it.
3) Statistical analysis: Please write in the methodology what logrank is used for in the research. In addition, the author seems to have used methods such as Cox regression, which need to be specified in the methodology.
4) Statistical analysis: The author used Pearson's correlation, and before using this method, the author also needs to indicate whether the data meets the usage conditions.
5) Primer correctness: Please ask the author to verify if the primers are correct.

Validity of the findings

1) The main text data needs to correspond one-to-one with the caption, for example, fig1b is a survival curve, and the author has no description of the results of the survival curve, and the content in the main text does not seem to correspond.
2) It is suggested to keep the image consistent with the article, for example, fig2a "mesenchymal epithelial transition (MET)" does not match the title in the image.
3) There is ambiguity in the representation of significance symbols in this article, such as fig2b. Which groups are compared with the significance symbols corresponding to each gene? Please indicate.
4) The main text of fig3a describes the discovery of 9 related genes through lasso regression, but it seems that the fig3a images do not show 9. Please provide higher definition images.
5) Suggest adding standard lines to make the results more intuitive, while also adding p-values to make the results more convincing.
6) Please indicate clearly which group of fig6c is high risk and which group is low risk. At the same time, the author needs to check whether the graphic and textual descriptions correspond.
7) Fig7a seems to have 11 cell types, please check the text.
8) Cell blurriness: fig8b cells appear blurry with unnatural traces. Please provide the original data.
9) Image clarity: The overall image clarity is poor, and some text cannot be seen clearly. High definition images are required, and the image text should be clear and visible.
10) Sample size: The sample size should be noted in the caption, such as fig8.
11) Supplementary experiment: The title and abstract of this article emphasize that three key genes were identified in this study, but only the METTL3 gene was included in the validation experiment. Due to the extensive research on METTL3 gene in colon cancer, the author only knocked out the METTL3 gene for migration and invasion experiments, and this result seems to have reached a consensus. Therefore, conducting such validation alone may not be innovative. Please explain why METTL3 was chosen.
(12) Unclear conclusion: The main conclusion is quite lengthy and does not summarize all the results. Please revise.

---

## Round 0.2 · accepted · Accept

Your revised manuscript has been thoroughly evaluated. Both reviewers have found that all their previous concerns have been satisfactorily addressed, and the modifications have significantly enhanced the manuscript's quality. Based on these positive assessments, I would accept your manuscript for publication.

Reviewer 1 ·

Basic reporting

no comment

Experimental design

no comment

Validity of the findings

no comment

Additional comments

Thank you for the editor's invitation again. I have carefully read the manuscript and the author has resolved all my questions. In this study, the author collected 21 m6A regulatory factors and explored their relationship with the progression of colorectal cancer (CRC). Then, based on significant prognostic m6A regulatory factors and Cox regression analysis algorithm, a RiskScore model was constructed for risk stratification, prognosis, and immune infiltration assessment. This model showed certain advantages in predicting patient survival probability, and this study has potential significance for improving clinical treatment decisions for CRC.

Reviewer 2 ·

Basic reporting

The gene screening process in this study is experimentally rich, logically clear, and persuasive. In the verification section, the author made many modifications based on my suggestions, which basically solved my problem. Congratulations to the author for the full improvement of the manuscript.

Experimental design

no comment

Validity of the findings

no comment